# Seasonal variation and group size affect movement patterns of two pelagic dolphin species (*Lagenorhynchus obscurus* and *Delphinus delphis*)

**Silvana Laura Dans**[1,2,3]*, **Elvio Agustin Luzenti**[2], **Mariano Alberto Coscarella**[1,2], **Rocio Joo**[4], **Mariana Degrati**[1,2], **Nadia Soledad Curcio**[5]

**1** Centro para el Estudio de Sistemas Marinos, Centro Científico Tecnológico Conicet Centro Nacional Patagónico, Puerto Madryn, Chubut, Argentina, **2** Universidad Nacional de la Patagonia San Juan Bosco–Facultad de Ciencias Naturales y Ciencias de la Salud, Puerto Madryn, Chubut, Argentina, **3** Fundación de Historia Natural Félix de Azara, Centro de Ciencias Naturales Ambientales y Antropológicas, Universidad Maimónides, Ciudad Autónoma de Buenos Aires, Argentina, **4** Global Fishing Watch, Washington, DC, United States of America, **5** Instituto de Investigación y Desarrollo Pesquero, Mar del Plata, Argentina

* dans@cenpat-conicet.gob.ar

**Data Availability Statement:** All data and Rscript files are available from the github database (https://doi.org/10.5281/zenodo.7117362).

## Abstract

Movement is a key factor in the survival and reproduction of most organisms with important links to bioenergetics and population dynamics. Animals use movement strategies that minimize the costs of locating resources, maximizing energy gains. Effectiveness of these strategies depends on the spatial distribution, variability and predictability of resources. The study of fine-scale movement of small cetaceans in the pelagic domain is limited, in part because of the logistical difficulties associated with tagging and tracking them. Here we describe and model the fine-scale movement patterns of two pelagic dolphin species using georeferenced movement and behavioral data obtained by tracking dolphin groups on board small vessels. Movement patterns differed by species, group sizes and seasons. Dusky dolphin groups moved shorter distances when feeding and longer distances when traveling whereas the common dolphin did the same only when they moved in large groups. In summer, both dolphins cover longer distances in a more linear path, while in winter the movement is more erratic and moving shorter distances. Both species of dolphins prey on small pelagic fishes, which are patchily distributed and show seasonal variability in school sizes and distribution. However, dusky dolphins rely on anchovy to a larger extent than common dolphins. In Nuevo Gulf, anchovy shoals are smaller and separated by shorter distances in winter and dusky dolphins´ movement pattern is consistent with this. Dusky and common dolphins are impacted by tourism and fisheries. Further modelling of movement could be inform spatial based management tools.

## Introduction

Movement is a key factor in the survival and reproduction of most organisms ([1, 2]). Why an animal moves depends on its internal state and needs (searching for food and feeding,

**Funding:** Consejo Nacional de Investigaciones Cientificas y Tecnicas CONICET (PID 320/99, PIP 2015-2017 N˚112 201501 00615CO); Agencia Nacional de Promoción Científica y Tecnológica (Fondo para la Investigación Científica y Tecnológica FONCyT PICT N° 01 4030, 11679, 33934 and 2945, Empretecno FONARSEC N˚038); Fundación BBVA BIOCON 04 (2005–2008) and BIOCON 08 (2009–2012); PNUD ARG 02/018 (B B27); Fundación Vida Silvestre Argentina, The Mohamed Bin Zayed species Conservation Fund (2011–2014).

**Competing interests:** The authors have declared that no competing interests exist.

searching a mate, escaping from a predator) [3]. When searching for resources, we expect animals to use movement strategies that minimize the costs of locating them, maximizing energy gains [4, 5], and the effectiveness of these strategies largely depends on the spatial distribution, variability and predictability of resources [6].

The movement of predators mainly depends on the distribution of prey species, and its study provides an avenue for exploring bioenergetics and population dynamics. It is also affected by environmental variability, including interactions with other species and human activities. During the last decades, the study of movement of predators received great impetus through the miniaturization of tracking devices and the advent of satellite tracking, enabling the acquisition of large amounts of data over different temporal and spatial scales for terrestrial [7] and aquatic animals [8]. These technological advances are bringing paradigm-changing opportunities for ecological research [9], with important implications for the conservation and management of marine megafauna.

Despite these advances, the study of fine-scale movement of small cetaceans in the pelagic domain has received little attention, in part due to logistical challenges and ethical considerations of attaching telemetry devices to wild animals [10, 11]. Attaching tracking devices to dolphins that yield information on a temporal scale longer than in the order of days requires the capture and immobilization of the animals, and the devices are pinned to the dorsal fin or anchored to the blubber or muscle tissue. The long-term effects of tagging on health and reproductive success of wild cetaceans remains largely unstudied. These devices may also induce a metabolic cost on the energy balance of swimming dolphins [12]. Non-invasive techniques such as the identification of individuals based on natural marks and re-sighted thereafter (for example by capture and recapture techniques by photoID) may contribute to the study of cetaceans movement. However, these techniques have only been applied to the study of movement at large to medium spatial scale of species that have natural visible natural marks, such as the study of migratory movements of humpback whales [13], home-range of bottlenose dolphins [14], or long-distance movement of a lone common dolphin [15].

A further complexity in studying movement patterns of pelagic small cetaceans, is related to the unpredictable and dynamic nature of the distribution of their prey biomass across the seascape, or their prey field. The main food resources of pelagic dolphins are small pelagic fishes (e.g. herring, sardines, anchovies and mackerel), which are patchily distributed, constantly moving, and whose location may fluctuate annually, seasonally or even within shorter-time scales [16, 17]. These food patches are organized in hierarchical spatial structures, where patches at small scales are nested within patches at larger scales [18]. At the smallest scale, individuals are congregated into schools and swarms. Schools and swarms are typically congregated into patches linked to mesoscale oceanographic features. Finally, these patches are typically congregated within large scale areas that reflect the habitat boundaries [19]. For example it was proposed that schools (ms to 10s m) and clusters of schools (kms) of the Peruvian anchovy or anchoveta *Engraulis ringens*, are impacted by oceanographic processes occurring at spatial mesoscale (10s km) and sub-mesoscale (100s m to km) such as sharp fronts, filaments, eddies and internal waves [20] 2008. Therefore a forager should track such spatial distribution of prey toward the small-scale end of the system. Within a large-scale patch, a forager should use long travel distances and low turning frequency in order to find medium-scale patches. Once a medium-scale patch is found, the predator should search for small-scale patches by increasing its turning rate and reducing its travel distances. Consequently, the forager will reduce the area to be searched and the risk of moving out of the present patch [18]. Fauchald et al. [21] described 3 levels of patchiness of murres Uria spp. and capelin *Mallotus villosus* in the Barents Sea. At a large scale, they found similar and overlapping patches of capelin and murres with a characteristic scale of more than 300 km. Within these large-scale

patches they found similar and overlapping medium-scale patches with a characteristic scale of about 50 km. Finally, within the medium-scale patches they found similar but non-overlapping patches with a characteristic scale of a few kilometers.

Small cetaceans preying on small pelagic fishes are to show scale-dependent movement patterns that match the spatial structure of their prey field. Most small cetaceans do not migrate, and may perform the whole set of activities necessary for life within the relatively same general area, foraging and breeding in the same ground [22, 23]. Therefore, their movements would be mostly motivated by searching, finding and handling prey. We also expect that the hierarchical spatial scale structure of food patches, would be reflected by their fine-scale movement pattern.

Georeferenced focal group or individual tracking onboard small vessels represent an alternative way to yield valuable information on movement patterns at fine- and meso-scales. This is a common protocol in behavioral studies of dolphins, by which the same group or individual is followed during a period of time, and the behavioral states are directly observed and recorded continuously at regular time intervals [24–26]. Modelling the movement pattern at individual and group level can inform how these predators acquire resources in a heterogenous and dynamic environment. In addition, as behavior is directly observed, it is possible to relate movement and behavior instead of inferring it from movement. In this study we did focal follows and modelled the movement pattern of dolphins inhabiting the pelagic domain to gain insight into their fine-and meso-scale foraging strategies. We applied this approach to two species in two different ecosystems, dusky dolphins (*Lagenorhynchus obscururs*) inhabiting Nuevo Gulf and common dolphins (*Delphinus delphis*) inhabiting San Matias Gulf, in northern Patagonia, Argentina.

Dusky and common dolphins in Patagonia may be described as sympatric [27], since they overlap their trophic and environmental (physical features) niches to some extent [28, 29]. They prey mainly on small pelagic fishes, but also take some groundfish and squid [29, 30]. They are highly gregarious, living in groups in a fission-fusion social system (individuals may split or join constantly) possibly related with prey abundance and predictability. These dynamics lead to the formation of groups of variable size with different functions, e.g. larger groups traveling, and smaller groups in almost motionless activities like resting or milling, and also according to seasonal changes of prey distribution.

Both species are subjected to by-catch in trawling nets by fisheries [31]. They are also the focus of an expanding whale watching industry, which may alter the dolphins' normal activity patterns and/or the duration of feeding bouts [24], but long-term effects on habitat use and population vital rates are unknown. Therefore, understanding the spatial dynamics of these dolphins is important in order to provide science-based management advice.

There is evidence that these species move alternating feeding and traveling bouts, possibly associated with searching for food and catching and ingesting prey [22, 32–34]. The trajectories dolphins follow during these sequences of feeding and traveling bouts have not been studied in a context of movement ecology and search strategy. We expect that trajectories reflect dolphins' behavior between and inside prey patches, so that distances traveled between patches would be larger than when they are feeding within a patch. These distances would also depend on the density and the distances between prey patches [35]. Therefore, we expect seasonal variability in prey density and distribution to influence the distances travelled and trajectories of dolphins.

Our goal was to assess the fine- and meso- scale movement patterns of these two pelagic dolphins in Patagonia. We evaluated the effects of species, behaviour, season (as a proxy for variability in the prey field), and group size on the movement pattern and trajectories travelled.

## Material and methods

### Study area and species

The study subjects are dusky dolphins inhabiting Nuevo Gulf (42˚20' 42˚50' S; 64˚20' 65˚00' W) and common dolphins inhabiting San Matias Gulf (40˚50'S to 42˚15'S and 63˚05'W to 65˚10'W), Argentina, in the southwestern South Atlantic (Fig 1). Both gulfs are semi enclosed basins, located in the northern Patagonia coast, with some common oceanographic and physiographic features. They are separated from the Atlantic Ocean by a shallow sill, which limits the interchange of water masses between the gulf and the Atlantic. Maximum depths of the basins are located in the central area and exceed 150 m. The major stratification in temperature as well as in salinity takes place in the spring-summer (September to March) being the warmer period from December and March, with several upwelling processes driven by westerly winds [36, 37].

Nuevo Gulf is an oval shaped gulf which is approximately 2500 km2 and its maximum depth is 184 m, connected to the Atlantic Ocean through a shallow sill of 16 km (Fig 1). Surveys were performed on the west side of the gulf, covering 805 km$^2$ (Fig 1). Here, the dominant westerly winds produce a sea circulation pattern characterized by an anti-cyclonic gyre at the west side and a cyclonic gyre at the center east [38]. The resulting circulation is sensitive to variations in topography [39]. The zone is also under the influence of wind and tidal currents and periodically receives intrusions of water from outside the gulf that increase the environmental variability, especially in the south coast [40].

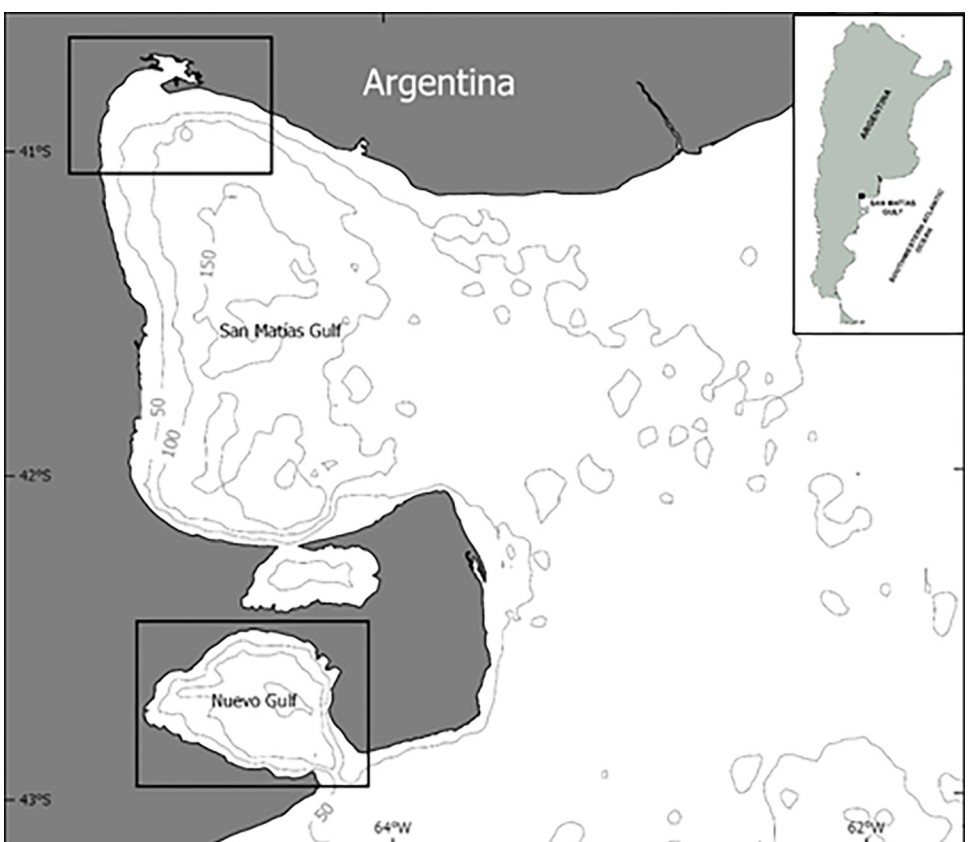

**Fig 1. Map showing the location of study sites.** The squares indicate the areas where surveys were carried out. Isobaths indicates depth in meters.

The number of dusky dolphins using this area was recently estimated as 374 (95% CI: 237–511) dolphins [41]. Population structure and connectivity between dusky dolphin groups using Nuevo gulf and open waters are poorly known. However, available information suggests a unique population in the Southwestern south Atlantic [42]. Dolphins use areas between 3 and 5 km from the shore and depths between 30 and 70 m [43].

San Matias Gulf is a larger basin than Nuevo Gulf, covering an area of approximately 20,000 km2, around 55% of the total area is deeper than 100 m, with a maximum of 180 m in the central area, and a mean depth of 70 m at the entrance [37]. During austral summer (from November to March), the northern and western area has high temperature and salinity, a marked thermocline, limiting nitrate concentrations and lower renewal rate than the southern and south-eastern area (strongly influenced by the intrusion of cold water from outside). In the San Matias Gulf, we focused surveys at the northwest area, based on known dolphins occurrence [28], and due to logistical constrains. The study area covered 930 km$^2$, near San Antonio bay (Fig 1). During the warm season, this particular area presents high values of sea surface temperature (SST) and periodically marked gradients of chlorophyll a (Chla) and SST [44], is also under the influence of a cyclonic circulation pattern [36] and seasonal wind driven upwelling that enrich the water with nutrients [45].

In the study area in San Matias Gulf, common dolphins and dusky dolphins co-occur throughout the year and occasionally form mixed species groups [28]. However, common dolphins are more abundant and more frequently found in a coastal strip closer to the coastline than dusky dolphins. In the present study, we only included common dolphins, since the small sample size of dusky dolphin groups found in the selected study area, precluded us to include them in the analysis.

## Sampling protocol and data source

Dolphin groups were tracked by 6–8 m boats with outboard engines from 2001 to 2017 in Nuevo Gulf (dusky dolphins), and from 2014 to 2017 in San Matias Gulf (common dolphins) (Fig 1). Surveys covered all seasons, although sampling periods were larger in summer months due to weather and sailing conditions (Table 1). Surveys consisted of non-systematic transects

**Table 1. Survey effort (days), number of dolphin groups and mean number of locations per group, along the months, for each dolphin species.** The values in the total row correspond to the sum of monthly values of days of survey effort and number of groups, and the Grand mean of the number of locations per group. Locations correspond to the position recorded every 2 min in a handheld GPS during groups tracking.

| Month | Common dolphin | | | Dusky dolphin | | |
|---|---|---|---|---|---|---|
| | Survey effort (days) | Number of groups | Mean number of locations | Survey effort (days) | Number of groups | Mean number of locations |
| January | 2 | 8 | 16 | 31 | 81 | 12 |
| February | 1 | 5 | 14 | 35 | 88 | 17 |
| March | 1 | 4 | 18 | 25 | 66 | 20 |
| April | 2 | 7 | 17 | 14 | 26 | 24 |
| May | 1 | 2 | 29 | 2 | 2 | 27 |
| June | 1 | 6 | 19 | 4 | 5 | 25 |
| July | 2 | 4 | 31 | 3 | 7 | 20 |
| August | 1 | 1 | 36 | 6 | 9 | 15 |
| September | 4 | 14 | 8 | 5 | 9 | 26 |
| October | 3 | 14 | 10 | 12 | 22 | 23 |
| November | 2 | 4 | 10 | 7 | 13 | 16 |
| December | 1 | 2 | 6 | 3 | 3 | 41 |
| Total | 21 | 71 | 14 | 147 | 331 | 18 |

throughout the study area, until a group of dolphins was detected. At this moment, the group was approached slowly, from the side and rear, with the vessel moving in the same direction as the animals and at a minimum distance of 100 m, in order to avoid any disturbance on behavior [24].

Each dolphin group approached by this way was defined as a focal group and followed as long as possible [46]. Observations were made at the group level instead of the individual level since dolphins move in groups, and identifying and tracking an individual dolphin within a group is very difficult without artificial or natural marks. In addition, the behavior of individual dolphins in a group is likely group-dependent [46]. Besides, groups are distinctive enough in the field, sometimes separated by several hundreds of meters from each other. This sampling methodology was already used for modeling habitat use of dusky dolphins in Nuevo Gulf [43] and for Chilean and Peale's dolphin movement pattern analysis [47]. It was also used for evaluating the effect of tour boats on dolphin behavior by modeling behavior sequences by Markov chains in dusky [24, 48] and common dolphins [26].

Once approached, the focal group was assigned to a size category (<10, 10–20, 20–50, 50–70, 70–100, >100; see [22] and the position was recorded by a handheld GPS on board. The GPS was set for saving positions every 2 min thereafter up to the ending (Fig 2). The behavioral state of individuals in the focal group was sampled by using an instantaneous sampling protocol [49, 50], recording the behavioral state at the beginning of the tracking sequence and every 2 min thereafter. Behavioral state was defined as the activity in which most members of the

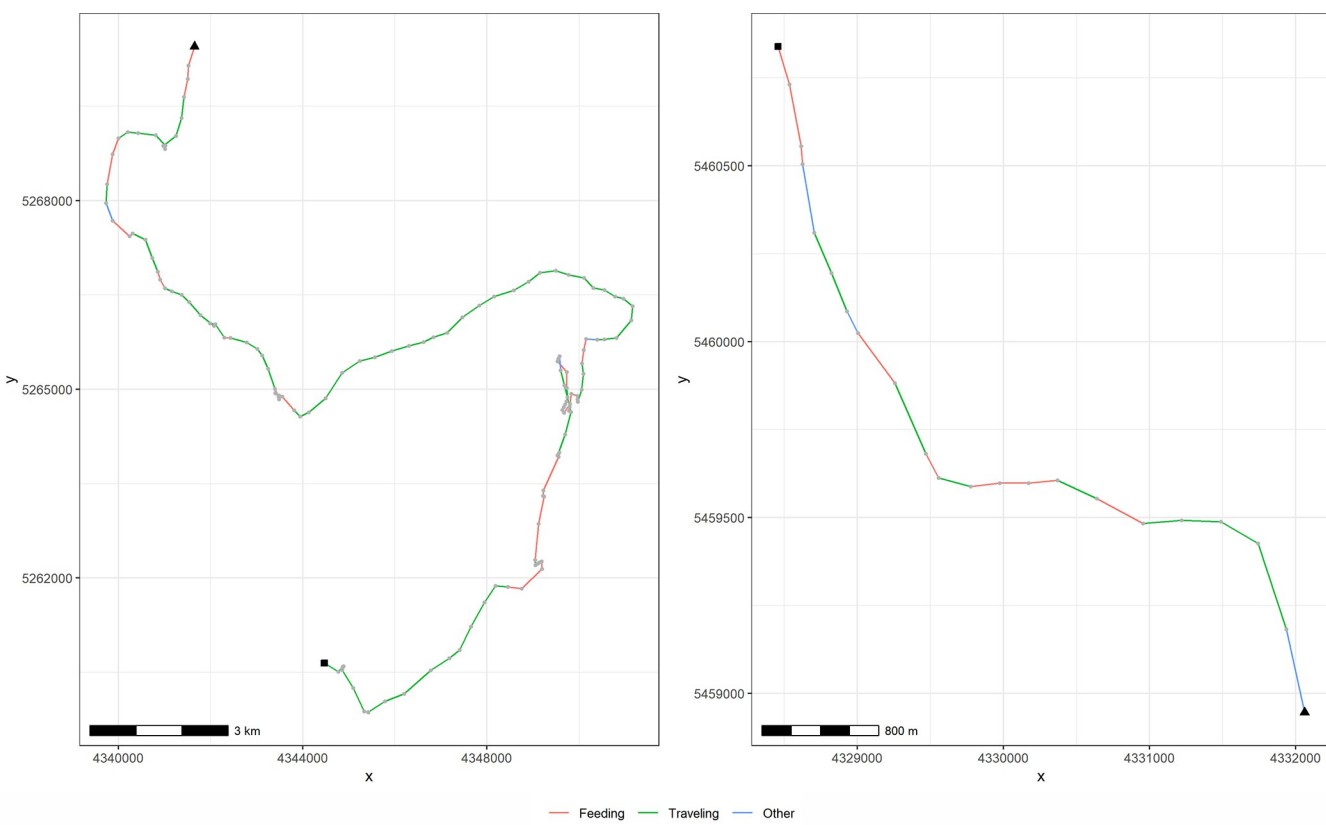

**Fig 2. Example of trajectories extracted from dolphin group follows by the package adehabitatLT.** The triangle and the square represent the beginning and the ending location respectively. Each grey circle corresponds to X and Y coordinates of each location recorded by the GPS on board. The "other" category includes milling, resting and socializing behavioral states. The left figure correspond to a group of dusky dolphins followed in Nuevo Gulf (09:57:00 14:31:00), and the right figure corresponds to a group of common dolphins followed in San Matías Gulf (16:31:18 17:35:18).

group were engaged: traveling (dolphins moving continuously in a single direction with few or no interruptions), feeding (dolphins moving fast, diving, and emerging in all directions, dolphins pursuing and chasing fish, fish jumping out of water, and marine birds feeding simultaneously), socializing (dolphins in almost constant physical contact with each other, belly to belly swimming, aerial displays frequently noisy such as leaps, tail-over-head leaps, backslaps, headslaps, and tailslaps), milling (dolphins moving slowly, changing direction continuously, and shifting location), and resting (dolphins tightly grouped, swimming slowly with numerous direction changes, and not shifting location) [22, 24, 48].

Since the position was recorded by the GPS automatically every 2 min interval, and the behavioral state was recorded in forms during the tracking, both data were matched a posteriori. To minimize the potential impact of the boat on the behavior of the study animals, boat behavior and distance to dolphins remained relatively constant at 100 m during the follow, and dolphins were approached from the side in the same direction and speed of dolphins' movement [24, 48]. Then this sampling protocol allowed that the trajectory of the boat mimic the trajectory of dolphins group. The survey finished if sea state was deteriorated by wind so following dolphins and assessing the behavioral state became difficult or imprecise (usually a Beaufort 4 or more, when whitecaps became very frequent).

## Data analysis and modeling

In this study a trajectory was defined as a set of consecutive point locations, where a group of dolphins were located at every 2-min, and a step was defined as the distance between the starting and ending locations of each 2 min interval recorded by the GPS. Trajectories were extracted from focal group trackings stored in the GPS. For each step $i$ we extracted the step length $l_i$ and the turning angle, $\theta_i$, defined as the angle of a step relative to the previous step direction [2] (S1 Fig). These metrics were calculated for each group trajectory, by using the package adehabitatLT [51] in R software version 4.0.2 [52]. Only tracks with a sufficient number of relocations to measure both a step length (i.e., requires two locations) and a turning angle (i.e., requires three locations) were included.

We modeled step length as a function of several explanatory variables considering that the species, the group size, the behavior, and the season may influence the movement pattern, by means of generalized linear mixed models GLMMs [53]. This modeling approach was chosen because several observations came from the same group followed, and hence, the group was used as the random effect term. Also, the distance covered in a time step is likely correlated to the distance covered in the preceding step(s), because they may be associated to a same goal (e.g., maintaining similar traveling speed when moving towards a prey patch), responding to similar environmental conditions, or simply due to inertia. For that reason, an autoregressive term was included in the model structure.

To study the relationship between step length and behavior, and the consistency of the pattern between species, we considered the species and behavior and their interaction as explanatory variables. As we were interested mainly in movement within a context of foraging, we differentiated between feeding and traveling steps, and the remaining behaviors were pooled within an "other behavior" category. Degrati and coauthors [22] found that feeding bouts last longer in larger groups, therefore the group size was also considered as a covariate, in order to control its potential effect. The inclusion of group size also allowed the evaluation of intraspecific variability.

In order to better describe the movement pattern, we included the turning angle as a covariate in the model. Turning angle provides a measure of persistence in a direction. If a dolphin group has a tendency to continue in the same direction as the previous step, the mean turning angle will be zero. Turning angle was measured in radians, so a turning angle of 0 means that

there was no variation in the direction between one step and the previous one, and it can vary between -π and π radians, depending if the change was clockwise or counterclockwise. As we were only interested in the straightness of the movement and not in the direction of its change, we used the cosine of turning angle. Since we expected longer and straight movement when dolphins were traveling, we included the interaction of the cosine of turning angle with the behavioral state.

To take into account possible environmental influences on movement,, and if these influences are consistent between species, the season and the interaction between species and season were also added as explanatory variables. In order to avoid very small sample sizes and convergence problems, data were pooled into Winter (from May to October) and Summer (November to April).

To find the best model structure, we compared the performance of the model with and without a random effect, and with two different error distributions. The full model incorporated the group ID as a random effect, a first order autoregressive component, all the covariates (species, turning angle, behavior, group size, season) and the whole set of double interaction terms (ten double interactions). We compared two possibilities for the error distribution, Gamma (with a log link function), and Gaussian (with identity link function). The Gaussian distribution acted as a null model to compared with the Gamma distribution, that would account for skewness in the tail distribution due to few long steps and many short steps [54] in dolphin movement. The parameters of the models were then estimated by restricted maximum likelihood REML [53]. All model comparison and selection was based on the Akaike information criterion AIC, the model with the lowest AIC was considered to be the best model [53, 55].

Once the best model structure was found, we compared nested models for hypothesis testing (following the protocol suggested by Zuur et al. 2009, p90 [53]). The full model included all double interaction terms and was estimated by maximum likelihood ML. Then each interaction term was dropped in turn, and each model with a dropped term was compared to the full model by evaluating the difference in AIC. Double interaction terms that did not improve the model were excluded, and a new full model was estimated. From this new full model, the remaining double interaction terms were evaluated in turn again in a second round, following the same protocol.

Previously to modelling, correlations between explanatory variables were computed to make sure that the values were low enough (under 0.5) to consider them as independent explanatory variables. For diagnosis we used the package DHARMa 0.3.2.0 [56]. All statistical and modeling analyses were made using the statistical software R version 4.0.2 [52]. For GLMMs we used the package Template Model Building glmmTMB [57]. We chose this package due to the possibility of fitting a gamma function. TMB is more flexible and faster compared to other classical packages like lme4 [57], and use a better approximation to the likelihood estimation (based on Laplace instead of penalized quasilikelihood) [58]. It also allowed for the computation of profile confidence intervals, based on likelihood ratio tests which are less sensitive to sample sizes than the Wald method which depends on normality of the ml [59].

This work was carried out under permits of the Dirección de Fauna y Flora Silvestre and Ministerio de Turismo de la Provincia de Chubut and complies with the current laws of Argentina.

## Results

A total of 402 different groups' trajectories were obtained, 71 for common and 331 for dusky dolphin. From this total, it was possible to calculate the step length for 385 trajectories (7

**Table 2. Mean step length and standard deviation in m, in each behavioral state for common and dusky dolphins.**

| Behavioral State | Mean step length (SD) | | Number of steps | |
|---|---|---|---|---|
| | Common dolphin | Dusky dolphin | Common dolphin | Dusky dolphin |
| Feeding | 114.11 | 161.86 | 140 | 915 |
| | (106.19) | (144.35) | | |
| Traveling | 197.75 | 274.62 | 403 | 2062 |
| | (102.78) | (162.85) | | |
| Other | 131.94 | 153.06 | 403 | 2623 |
| | (85.86) | (124.70) | | |
| Total | 199.24 | 157.34 | 946 | 5600 |
| | (154.16) | (102.65) | | |

groups had only 1 relocation), 71 for common and 314 for dusky dolphin (Table 2), and the turning angle for 368 trajectories, 66 for common and 302 for dusky dolphin.

Both species showed skewed step length distributions, although dusky dolphins showed a more heavily tailed distribution (Fig 3). It was possible to calculate the turning angle for 6161 steps (875 for common and 5286 for dusky dolphins). Turning angle did not show a uniform distribution and rather a symmetric distribution centered at zero, suggesting a persistence in movement direction in both species (Fig 4). Dusky dolphins showed the maximum step length of 1551.1 m, while common dolphins showed a maximum step length of 650 m; in both cases larger groups moved larger distances, but this trend was more noticeable in common dolphins (Fig 5), and maximum values of step length were recorded when dolphins were traveling (Fig 6). The relationship between turning angle and behavior is less evident but values are more concentrated around 0 when dolphins are traveling than when they are feeding (Fig 7). A seasonal variation in step length was evident in dusky dolphins, with shorter step lengths during winter months, while there was no clear pattern in common dolphins (Fig 8).

The best model for step length had a Gamma function for the error. The incorporation of a random effect term and temporal autocorrelation provided a lower AIC compared to that of a model without these terms both for a Gamma and a Gaussian function (S1 Table in S1 File).

When testing for hypotheses represented by the fixed effect terms, six interactions were removed. The final model contained main fixed effects, since the whole set of explanatory variables initially considered were included, and also some interactions between them: Behavior*Species, Group size*Species, Behavior*Group size and Turning angle* Season (Fig 9) (Table 3) (regression coefficients for interaction terms as well main effects are shown in S2 Table in S1 File; see S2 Fig for residuals inspection).

Among the six removed interaction terms, the interaction Turning angle*Species was considered marginally significant since the difference in AIC was 2.1; this interaction term does not have a biologically important interpretation, therefore it was excluded from the final model.

There was a significant relationship between step length and behavior; dolphins performed longer steps when they were traveling than when they were feeding or doing something else. However, the relationship is different depending on the species, since the interaction term was significant (Behavior*Species in Table 3). The difference between the mean step length when dolphins are traveling and feeding, is more pronounced in dusky than in common dolphins, since predicted curves for feeding and traveling step length are less overlapped in the dusky dolphin (Fig 9).

Group size had an important effect on step length, since two interaction terms including the group size significantly improved the model. In general, larger groups performed longer

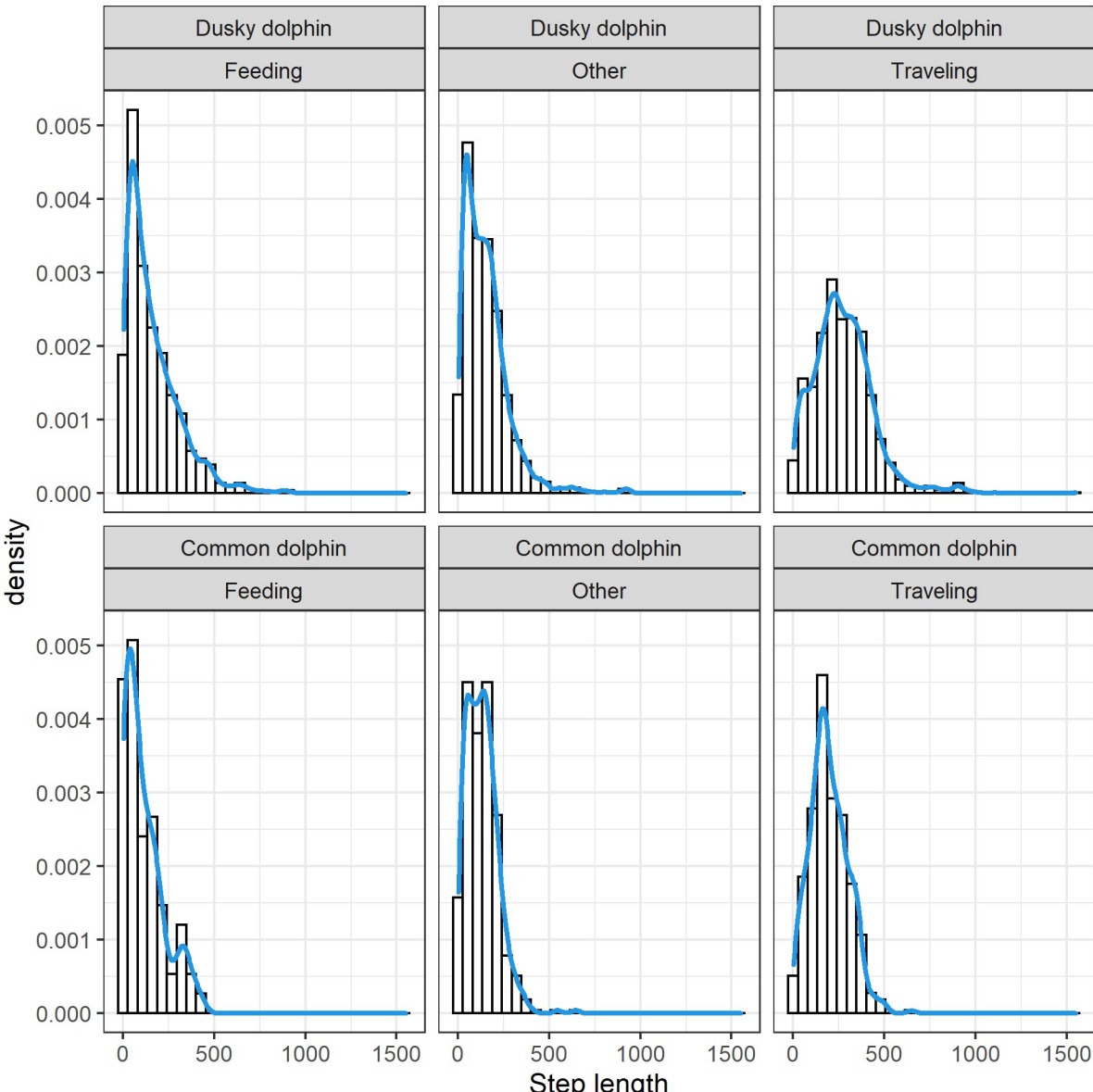

**Fig 3.** Step length relative frequency distribution (histogram and density curve) for dusky (upper) and common dolphin (lower) groups' trajectories for each behavioral state: Feeding, Traveling, and Other. The "other" category includes milling, resting and socializing behavioral states. Each step corresponds to a 2 min interval within focal group follows, and its length is measured in meters.

steps, but the relationship depends on the species and behavior, since these interaction terms were both significant (Group size*Species and Group size*Behavior in Table 3). Larger dusky dolphin groups did longer steps, as described previously, but steps length is quite similar among different group sizes, compared to common dolphins—The predicted curves of step length are more separated among different common dolphin group sizes, and more over-lapped among dusky dolphin group sizes (Fig 9). This tendency to perform larger steps when groups are larger also depends on the behavior; particularly larger groups perform longer steps when they are traveling than when they are feeding. Although the 3-ways interaction term was not tested, predicted curves for dusky dolphin group sizes are more overlapped, although different between feeding and traveling. In the case of common dolphin, curves are more

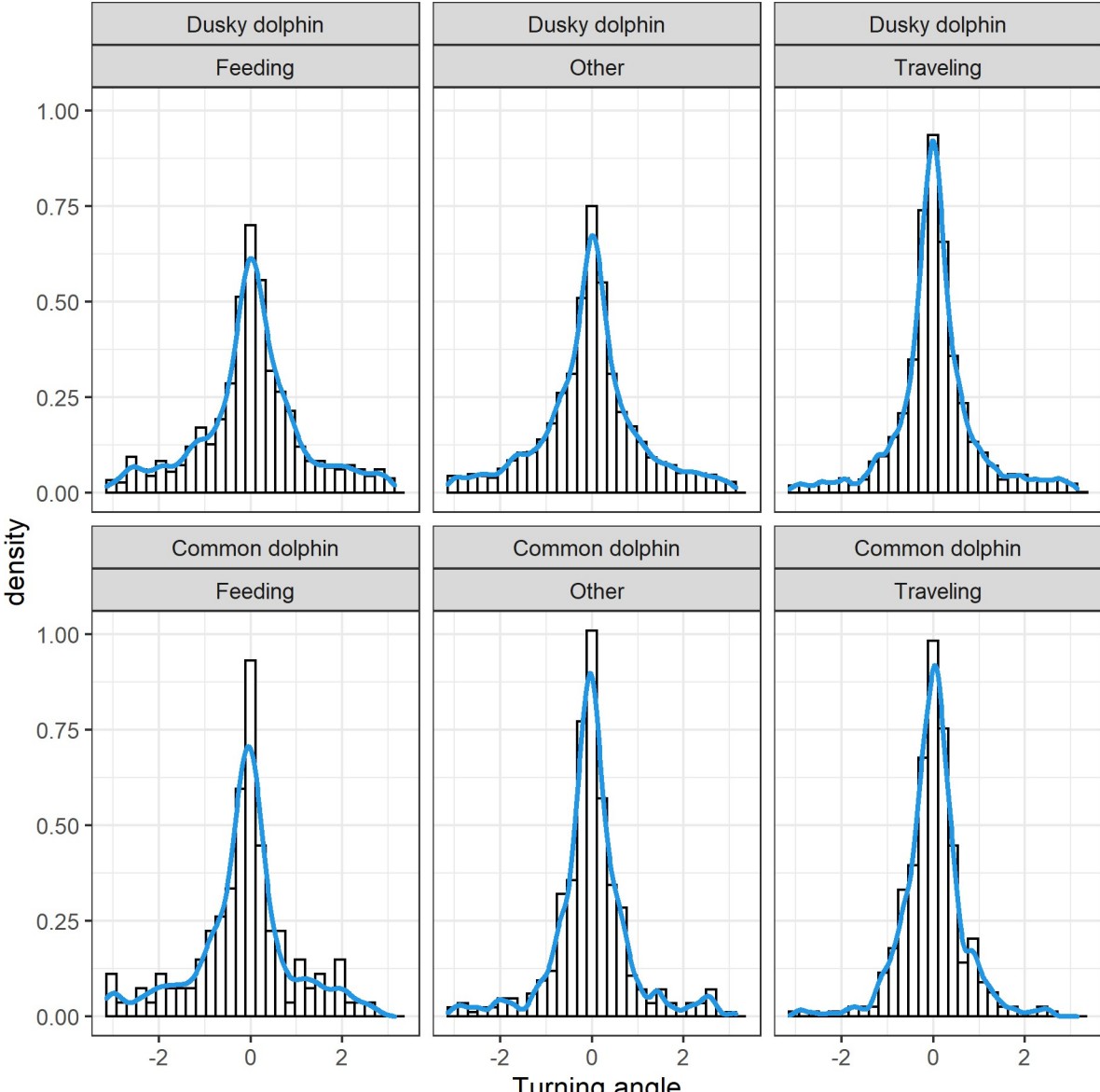

**Fig 4.** Turning angle relative frequency distribution (histogram and density curve) for dusky (upper) and common dolphin (lower) groups' trajectories for each different behavioral state: Feeding, Traveling, and Other. The "other" category includes milling, resting and socializing behavioral states. Turning angle is measured in radians, so a relative angle of 0 means that there was no variation in the direction between the current and the previous step.

separated among group sizes, then the degree of overlapping comparing feeding and traveling, is higher. However, the curves do not overlap looking at larger common dolphin groups, and this is more evident for the last size category which is >100 animals (Fig 9).

Seasons had an important effect on the movement pattern (Fig 9), and the seasonal change could be associated with the straightness of the path. Season seems to have the largest effect on the relationship between step length and turning angle. When interaction terms were dropped in a second round from the final model, the exclusion of the interaction term Turning angle* Season showed the major difference in AIC with respect to the final model, compared to the other interaction terms (Table 3). The seasonal variation of the relationship between turning

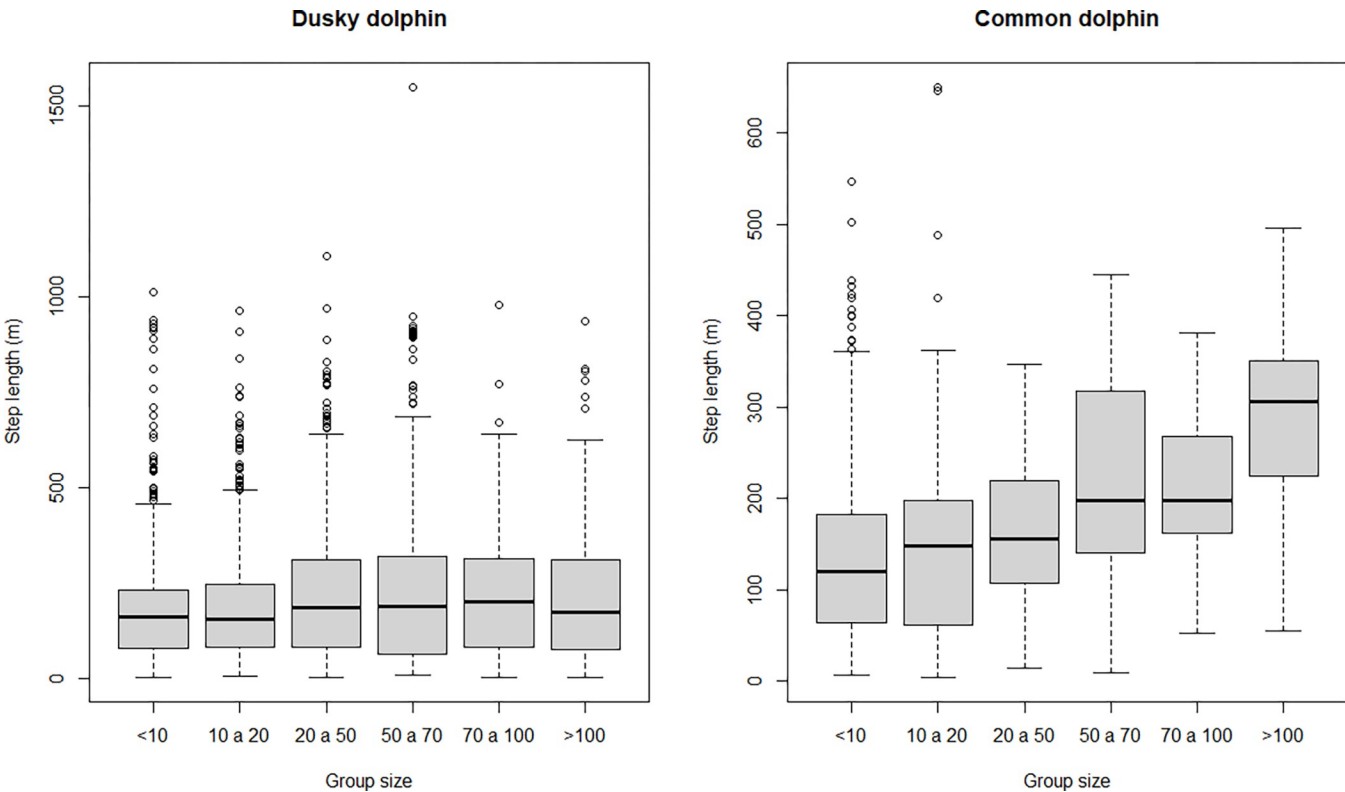

**Fig 5. Relationship between step length and group sizes for dusky and common dolphins.**

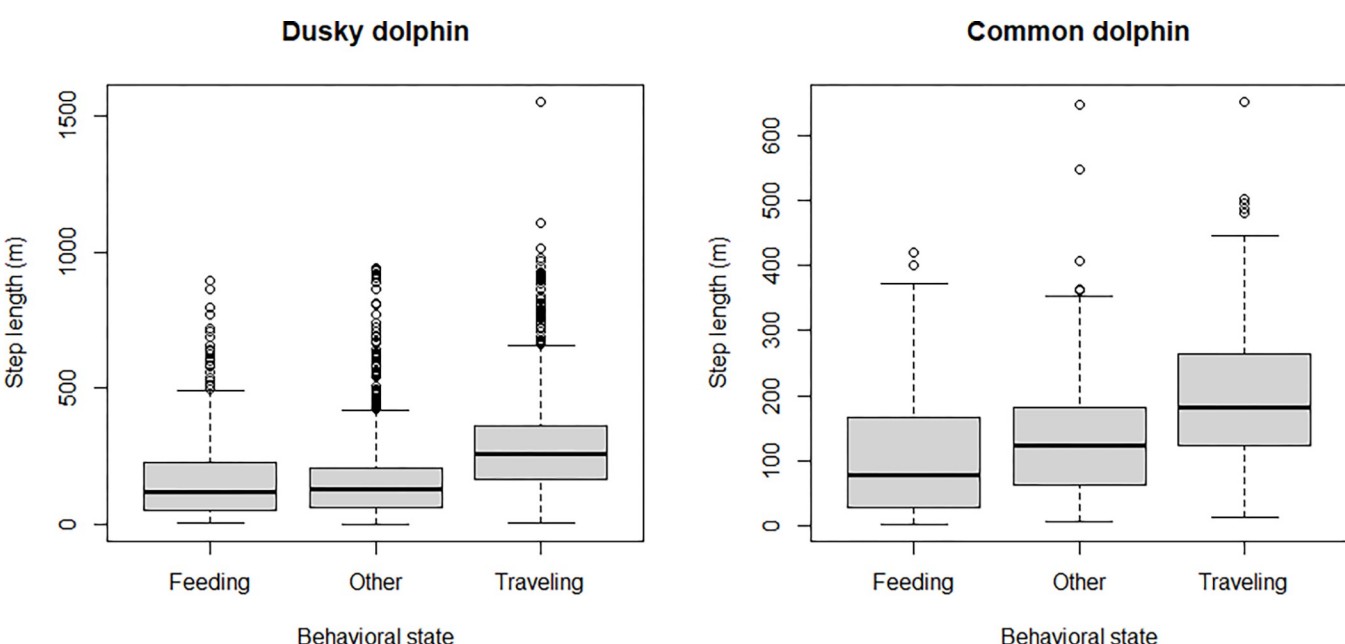

**Fig 6. Relationship between step length and behavioral state for dusky and common dolphin groups trajectories.** The "other" category includes milling, resting and socializing behavioral states.

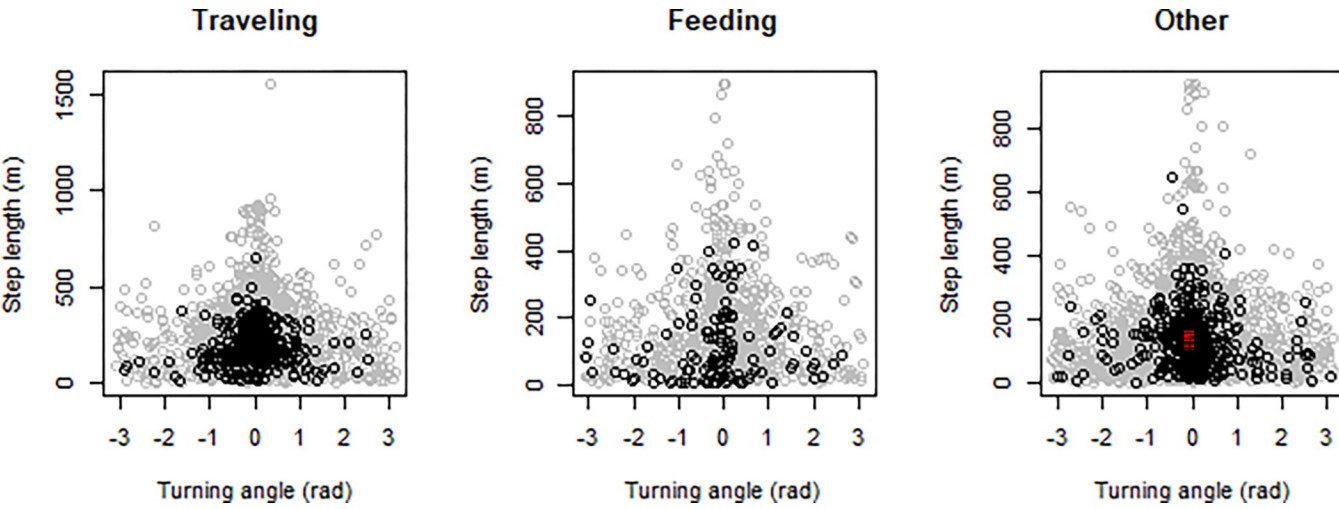

**Fig 7. Relationship between turning angle and step length in each behavior, for common (blue) and dusky dolphin (orange) groups' trajectories.** Turning angle is measured in radians, then an angle of 0 means that there was no variation in the direction between one step and the previous step. Step length is measured in meters. The "other" category includes milling, resting and socializing behavioral states.

angle and step length, is related to the height of the curve of predicted values (Fig 9), being greater in summer, and meaning that dolphins move longer distances in each step and in a straighter way in summer. The apparent seasonal pattern in step length observed for the dusky

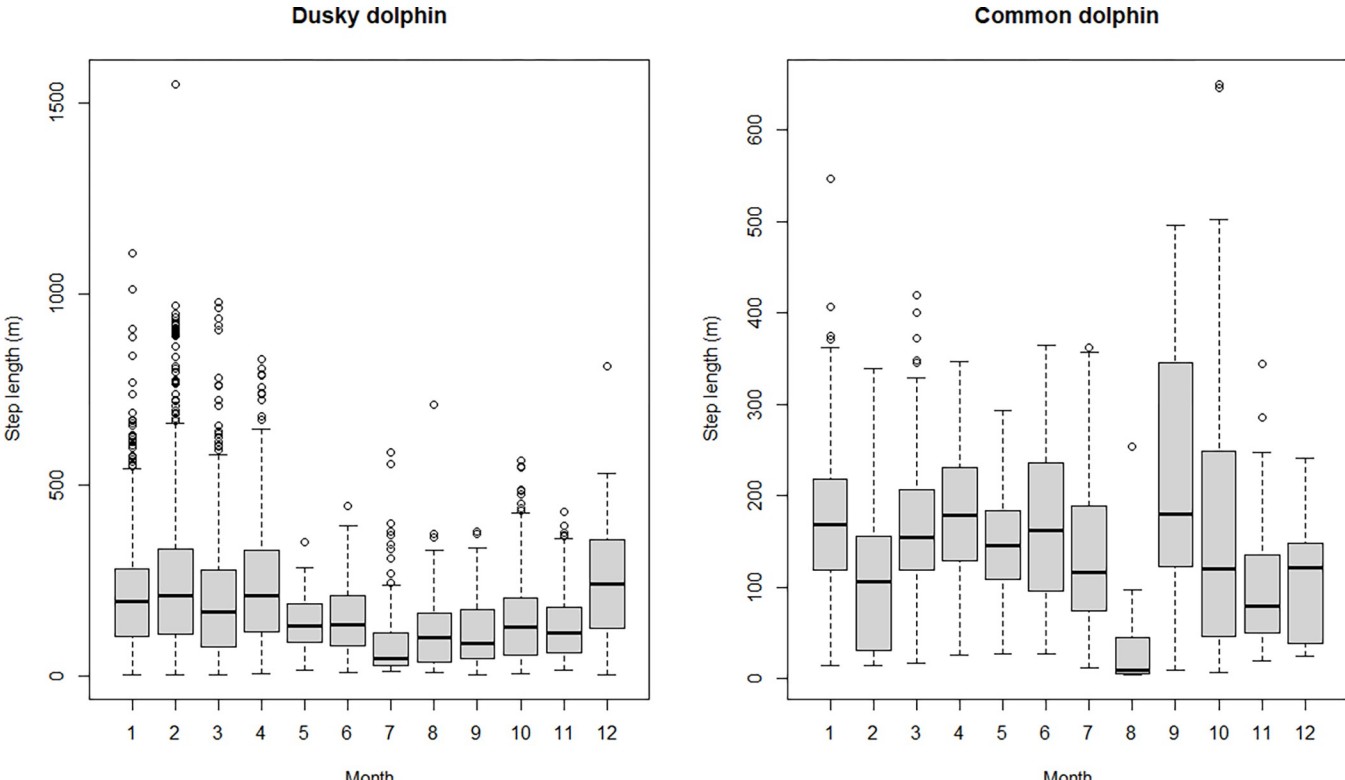

**Fig 8. Relationship between step length and month of the year for dusky and common dolphin groups trajectories.** Months were pooled in "Summer" (November to April) and "Winter" (May to October) for incorporating season as a covariate in subsequent analysis.

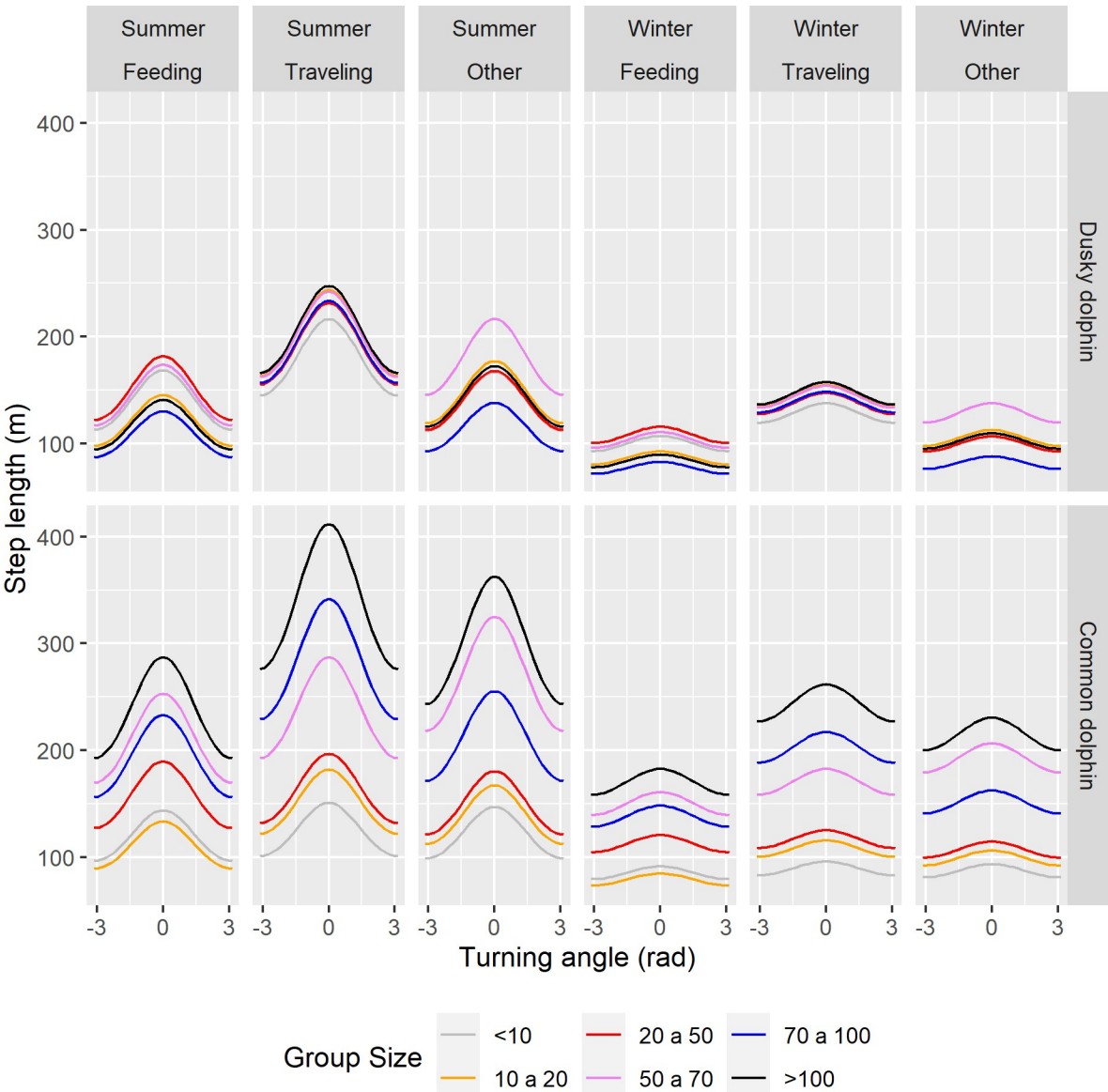

**Fig 9. Mean Step length predicted by the selected best model, which includes a gamma function, a random effect term, temporal autocorrelation, and four double interactions: Species\*Behavior, Turning angle\*Season, Group size\*Behavior and Group size\*Species.** Group size categories in different colors. The "other" category includes milling, resting and socializing behavioral states.

dolphin (see Fig 9), was not captured by the incorporation of the interaction term effect on step length into the model ($\Delta AIC = 1$, Species\*Season in Table 3). Therefore dolphin groups of both species did longer steps in summer and shorter steps in winter.

## Discussion

The present study is a first approach to analyze the movement pattern of two species of dolphins inhabiting the pelagic domain. Behavioral sequences consisting of traveling and feeding bouts have been already described for the dusky dolphin in Nuevo Gulf [22, 24] and for the common dolphin in New Zealand [26], but never described for common dolphins in Patagonia, Argentina. This is the first time that these sequences obtained from group follows are

**Table 3. Model evaluation and selection of fixed effects terms.** The full model included Turning angle, Behavior, Species, Season and Group size as covariates and the whole set of double interactions. ΔAIC correspond to the difference between each restricted model and corresponding full model. Bold text indicates significant interaction terms that remained in the final selected model. In a second round, each interaction term was dropped from the final selected model, and evaluated by ΔAIC as previously done.

| Hypothesis | Model Structure | Explanatory variables | ΔAIC | df |
|---|---|---|---|---|
| Full model | Gamma Random effect (group) Temporal autocorrelation | Turning angle, Behavior, Species, Season, Group size, Whole double interactions | 0 | 49 |
| Dropped term: | | | | |
| **Behavior*Species** | | | **9.2** | **47** |
| Turning angle*Group size | | | 6.8 | 44 |
| **Species*Group size** | | | **5.8** | **44** |
| Season*Group size | | | 1.5 | 44 |
| **Behavior*Group size** | | | **6.1** | **39** |
| **Turning angle*Species** | | | **2.1** | **48** |
| Turning angle*Behavior | | | 0.2 | 47 |
| Species*Season | | | 0 | 48 |
| Behavior*Season | | | 1 | 47 |
| **Turning angle*Season** | | | **4.7** | **48** |
| New full model | Gamma Random effect (group) Temporal autocorrelation | Turning angle, Behavior, Species, Season, Group size, Behavior*Species Species*Group size Behavior*Group size Turning angle*Season Turning angle*Species | 0 | 34 |
| Dropped term: | | | | |
| Turning angle*Species | | | 2.1 | 33 |
| **Species*Group size** | | | **4.9** | **24** |
| **Behavior*Group size** | | | **5.9** | **29** |
| **Behavior*Species** | | | **6.8** | **32** |
| **Turning angle*season** | | | **49** | **32** |

spatially analyzed in terms of the trajectory followed by dolphins, quantifying and modeling distances moved in each time interval. Our approach demonstrates the usefulness of these data in studying movement patterns in a movement ecological framework in species with poorly known searching strategies.

In the present work, we showed that the path these species followed when searching for food and feeding consisted of short and long moves, with a skewed distribution, which means that dolphins move short distances more often, intermingled with less frequent longer distances. The observed movement pattern is consistent with dolphins preying on very patchily prey field, where pelagic fishes are concentrated in shoals, which are also concentrated in some places driven by oceanographic features. Considering the pelagic fishery as another predator preying on anchovies, Joo and coworkers [60] showed that environmental and anchovy conditions do significantly shape fishermen spatial behavior.

We showed that the distance dolphins move in every step is related to their behavior. Movement studies typically use latent models (e.g. Hidden Markov models), each one with particular inherent assumptions, to infer behaviors from tracking data, since they are usually not directly accessible through the sole observation of the sequence of positions recorded by GPS or other position-logging artefacts, requiring groundtruthed datasets [61], simulations [62], or expert judgement [63] to help fit or validate these models. In our case, behavioral states were previously defined and recorded by observers on board, and not chosen in a way that they

would be *a priori* easily recognizable based on path geometry, but rather on the ecological knowledge of the species and direct observations. This allowed us to use the observed behaviors as covariates in the model.

We found some intra and interspecific differences among the two dolphin species. It is possible that different groups have group-specific searching strategies. One factor affecting the step length would be the group size, since the relationship between behavior and step length depends on the group size: when larger groups are traveling, they move longer distances in each step. The formation of larger groups may be responding to changes in the location and searching for new prey patches. Both species are highly gregarious, and although some kinship relationships may exist among some members, the formation of larger groups respond to the benefit of higher efficiency in finding, handling and capturing pelagic fishes [22]. In addition, it is possible that each group has a group-specific movement pattern. The inclusion of the group as a random term accounts for statistical considerations, like avoiding pseudo replication, but also supports this conclusion. The existence of group-specific movement patterns may be related to age and sex composition. In the field, it is feasible to recognize pairs of mothers with their calves, and individuals which are comparatively shorter and assumed to be juveniles [22]. However, these age and sex classes are intermingled in the same group very often, and the classification of a group according to their composition is subjected to a large error than the group size. Also, it is possibly that group size and composition are correlated, since among dusky dolphins, mothers with their calves were usually segregated in smaller groups [22]. For this reason, we only considered group size as an explanatory variable as the best suited variable to account for inter group variability, but it is possible that the effect of age and sex composition it is already included.

The effect of group size on step length depends on the species, and this result suggests some interspecific differences. Larger common dolphin groups travel longer distances in each step, while dusky dolphin groups of different sizes move in a similar fashion (Fig 9). The effect of behavior on step length also depends on the species; dusky dolphin groups move longer distances in each step when they are traveling, and shorter ones when they are feeding, but this difference is not so obvious in the common dolphin (Fig 9). These interspecific differences in the movement pattern, suggest these species are differentiating their searching and foraging strategies as a way of niche partition. Spinner dolphins *Stenella longirostris* in the Hawaiian Archipelago increase prey-finding abilities by forming large groups when traveling offshore to meet the nocturnally-rising deep scattering layer (DSL), and during the day, they form small parties to "fit" within small nearshore bays, and reduce predation risk by resting over light-colored sand [64, 65]. Therefore, group size and behavior interplay an important role and it must be taken into account in further studies of dolphins searching strategies.

About seasonal patterns, both dolphins moved with shorter steps during winter, and longer steps during summer. Also, the season has a significant effect on the step length depending on the turning angle (Fig 9). This means that when dolphins do not change the direction, which is indicated by values around 0 in the distribution of turning angles, they will move a longer distance. Therefore, in summer, dolphins move longer distances in a more straightforward movement, while in winter, the movement is more erratic and moving shorter distances. The seasonal variation of the relationship between step length and turning angle of dusky dolphins' movement in Nuevo Gulf may be explained in terms of the abundance and distribution of anchovy shoals. There is some evidence about a seasonal change in foraging behavior of dusky dolphins inhabiting Nuevo Gulf [33]. Dusky dolphins were observed in coordinated diving apparently in a feeding activity, contrasting to the surface feeding observed during summer. This change was suggested to be related to a change in the location of anchovy shoals, since both anchovies and dusky dolphins were found in deeper waters during the cold season [34].

Although these results come from only one year, they suggest a seasonal pattern in prey distribution. More recently, a study about abundance and distribution of pelagic ensembles, including anchovies and red lobsters *Munida gregaria*, carried out in both gulfs during 2016–2018, indicated seasonal changes in the depth, the morphology and the energetic density of anchovy shoals [66]. In Nuevo Gulf, during winter, the number of shoals per unit area and the energetic density are higher during winter than during summer. However, shoals are smaller and separated by shorter distances during winter than in summer [66]. However, the seasonal variation of movement pattern of common dolphins in San Matias Gulf is more difficult to explain. In San Matias Gulf, shoals were also smaller in winter, but the number of shoals and the energetic density was lower [66].

The study of movement patterns of pelagic dolphins' groups is difficult to approach due to logistic limitations and hence the study of movement in most pelagic dolphins is almost nonexistent. Here, we gathered data to reconstruct trajectories in a way alternative to the use of attached electronic devices. Tagging and tracking dolphins in the wild, usually, but not always, require the capture, manipulation and release of animals. This alternative method for gathering and analyzing movement data without invasive techniques, may help to reducing costs, ensuring animals' welfare at the same time, and being applied to small cetacean species for which the manipulation is risky and/or unsuccessful. Although it has the restriction of not being able to follow the animals all the time, species studied in the present work feed mainly during daylight hours therefore the observed movement pattern would represent most of their foraging strategy at a fine scale.

One important aspect to discuss is the spatial and time scale of the observed pattern, considering that our data come from short tracking, lasting some hours at most. This temporal scale possibly preclude generalizations about medium to long term movements of dusky and common dolphins, however we expect that the movement pattern within prey patches occur at the temporal and spatial scale of our study. Some long range movements were reported for both species, being several orders of magnitude higher than the distances tracked in our study (hundreds of kilometers versus hundreds of meters). However, these movement were reported in a longer term along several years (e.g. two male dusky dolphins tagged in San Jose Gulf, a location near to Nuevo Gulf, were resighted 800 km northwards [67], and a lone female common dolphin in the Mediterranean Sea [15]. In addition, no migration was described and abundance of both species seemed to be stable along the year within our study areas [32, 41], therefore we expect dolphins not to leave the area at least in the short term of days or inclusively months. On the other hand, the spatial scale at which we studied the movement pattern respond to the spatial scale at which fish schools are distributed. For example the dynamics of a herring school occurs at the spatial scale of 10s of meters, whereas the spatial distance of medium scale dynamics would be 10s to 100s of meters [16]. Therefore the scale of our study is well suited for studying the fine scale movement pattern of dolphins at the mesoscale spatial distribution of prey (e.g. dolphins would be moving from one school to another or from one patch to another, but all of them occurring within the boundaries of our study areas and inside the gulfs).

Studying movement at the mesoscale of prey, could be the base for more sophisticated models which will allow to estimate additional parameters. Although the duration of our group follows did not last long, several hours at best, they may be used in estimating parameters such as the area of restricted search and net displacement. Available studies about habitat use and predictability of dusky dolphins location [23] and the way tourism boats search for dolphins [68], suggest that dolphins possibly remain in the same food patch during one day or inclusively several days, before moving to another patch. This in turn could be of conservation interest, and to manage tourism and fisheries in order to reduce negative effects (such as disturbance on dolphin normal energy budget due to vessels approach, and fishery by-catch), by

minimizing the encounter probability of dolphins and vessels. If tourism or fishing activities also concentrate in these areas, they may co-occur with dolphins and possibly increase the probability of interactions. Therefore modelling dolphin movement in the short term, may help in the development of managing tools based on spatial models.

Further modelling of the tracking data presented in this work would allow a more deep insight of the movement pattern and searching strategy of these species. The observed distribution of step length and turning angle (Figs 3 and 4), suggests that the movement pattern may be assimilated to a correlated random walk CRW [69]. It is expected that the movement pattern and the tendency to dispersal, that is to change the location along the time instead of residing in the same, would be associated with the distribution and predictability of prey. The results found in the present study suggest that the combination of mostly short moves with some long moves, is in accordance with food patchily distributed, patches highly mobile and hence unpredictable to some extent. Such search tactic also suggests that dolphins will show a tendency to dispersal. This hypothesis remains untested but further modelling could be done in order to test it, developing more mechanistic models of movement—that allow for simulation of movement, for instance. The study of movement pattern, within a CRW framework, was applied to other two dolphin species, the Peale's and the Chilean dolphin, in southern Chile [47]. The lack of fit of these dolphin species to CRW models was interpreted as these species do not undertake movement patterns typical of dispersal, but rather have a tendency to reside in the same locations. Possibly the habitat use of these species is more restricted due to prey type, but also because the coastal environment they inhabit in physically limited fiords. Therefore the hypothesis of a correlated Random walk remained untested for pelagic dolphins.

The marine habitat is very dynamic, and environmental conditions may change spatially and temporally, and the behavior and location of predators will follow these changes. Therefore, tracking data must be coupled with environmental data, to detect the extent of areas of concentrated use as well as the stability or not in a particular location. Therefore future movement models should incorporate physical drivers, like temperature, depth, distance to the coastline, and biological drivers, like prey abundance, school sizes and school depth, in order to explain and predict dolphins' movement. As mentioned before, several acoustic surveys for studying pelagic assemblages (including anchovy) were done during 2016–2018 in both study areas (four surveys in San Matias Gulf, corresponding to one survey per season, and five surveys in Nuevo Gulf, one survey per season and replicating the winter survey [66]. However due to the very variable dynamics of pelagic fishes, we could not match these data to dolphin group follows, in order to add the prey as an explanatory variable. The incorporation of prey to movement models represents one goal to approach in the future.

Dusky and common dolphins are subjected to several impacts of human activities. How dolphins move may be used in predicting their location in very short time periods, and then the probability of encounter with human activities like touristic boats and fishing vessels. Further studies should focus on quantifying the extent of the area dolphins move, their degree of residency, as well the individual and temporal variation, which in turn may help in evaluating the impacts of these human activities or the detection of core areas (for example Area of Restricted Search) [70]. Our results and further models based on tracking dolphins could be particularly useful in spatial planning, where more spatially and temporally flexible actions are being required [71, 72].

## Supporting information

**S1 Fig. Descriptive parameters of a trajectory automatically computed by adehabitatLT in an object of class "ltraj": The basic unit of the trajectory is the step, the step length *li* is**

**defined by the increments in the X and Y directions for the starting and ending positions, and the turning angle θ*i* measures the angle between the current step and the direction of the previous step.**
(JPG)

**S2 Fig. Plot of residuals and predicted values from the final selected model.**
(TIF)

**S1 File.**
(DOCX)

# Acknowledgments

This research received logistic and institutional support from Centro Nacional Patagónico (CONICET), Universidad Nacional de la Patagonia San Juan Bosco (UNPSJB), Centro de Investigación Aplicada y Transferencia Tecnológica en Recursos Marinos "Almirante Storni" (CIMAS), Escuela Superior de Ciencias Marinas (ESCiMar), Universidad Nacional del Comahue. Fieldwork was also supported by Hydrosport SRL and Asociación Civil Cota Cero Buceo. We mainly acknowledge staff of the nautical service of CCT CONICET CENPAT, CIMAS, and J. Owen for nautical services and logistics, members of Marine Mammals Laboratory CENPAT, G. Garaffo, G. Svendsen, M. Arias, A. Ramirez for her help during field work, and Patricia Dell´aciprete (CESIMAR-CONICET) for her assistance in the use of adehabitatLT. The authors also thank Dr. Alejandro Buren (CONICET at Instituto Antártico Argentino) who provided constructive comments, writing and english revisión, that greatly improved our manuscript.

# Author Contributions

**Conceptualization:** Silvana Laura Dans, Mariano Alberto Coscarella.

**Data curation:** Silvana Laura Dans, Mariana Degrati, Nadia Soledad Curcio.

**Formal analysis:** Silvana Laura Dans, Elvio Agustin Luzenti, Mariano Alberto Coscarella, Rocio Joo.

**Investigation:** Silvana Laura Dans, Mariana Degrati, Nadia Soledad Curcio.

**Methodology:** Silvana Laura Dans, Elvio Agustin Luzenti, Mariano Alberto Coscarella, Rocio Joo.

**Project administration:** Silvana Laura Dans.

**Software:** Elvio Agustin Luzenti, Mariano Alberto Coscarella, Rocio Joo.

**Writing – original draft:** Silvana Laura Dans, Elvio Agustin Luzenti, Mariano Alberto Coscarella, Rocio Joo.

**Writing – review & editing:** Silvana Laura Dans, Mariana Degrati, Nadia Soledad Curcio.

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
