## [Decision Letter · Decision Letter 0]

12 Sep 2022

PONE-D-22-19932Seasonal variation and group size affect movement patterns of two pelagic dolphins (Lagenorhynchus obscurus and Delphinus delphis)PLOS ONE

Dear Dr. Dans,

Thank you for submitting your manuscript to PLOS ONE. After careful consideration, we feel that it has merit but does not fully meet PLOS ONE’s publication criteria as it currently stands. Therefore, we invite you to submit a revised version of the manuscript that addresses the points raised during the review process.

In addition to the specific questions and edits suggested by the two reviewers, please make sure to address these comments: Please make sure you address the reviewer requests for clarifications of the text, and for re-writes of certain sections. In Figure 9, the differences in step length become more pronounced for groups of 50 or more common dolphins. Is this the case?  Do the analyses support this assertion? If this pattern was not addressed, can you test it now?  Otherwise, why are only groups larger than 100 mentioned in the text?

In Table 1, what do the values of 14 and 18 indicate for the mean number of locations for common and dusky dolphins, respectively?  The sum of the monthly means / 12 gives other values.

Table 3 could be moved to the supplementary material section.

We look forward to receiving your revised manuscript.

Kind regards,

David Hyrenbach, Ph.D.

Academic Editor

PLOS ONE

Journal Requirements:

2. Please ensure to move your statement about permits from the acknowledgment section to the methods section of the manuscript: "This work was carried out under permits of the Dirección de Fauna y Flora Silvestre and Ministerio de Turismo de la Provincia de Chubut and complies with the current laws of Argentina.

4. Please expand the acronym “PNUD ARG” (as indicated in your financial disclosure) so that it states the name of your funders in full.

"NO authors have competing interests." 

7. We note that Figure 1 in your submission contain [map/satellite] images which may be copyrighted. All PLOS content is published under the Creative Commons Attribution License (CC BY 4.0), which means that the manuscript, images, and Supporting Information files will be freely available online, and any third party is permitted to access, download, copy, distribute, and use these materials in any way, even commercially, with proper attribution. For these reasons, we cannot publish previously copyrighted maps or satellite images created using proprietary data, such as Google software (Google Maps, Street View, and Earth). For more information, see our copyright guidelines: http://journals.plos.org/plosone/s/licenses-and-copyright.

a) You may seek permission from the original copyright holder of Figure 1 to publish the content specifically under the CC BY 4.0 license.  

8. Please ensure that you refer to Figure 5 in your text as, if accepted, production will need this reference to link the reader to the figure.

Reviewers' comments:

Reviewer's Responses to Questions

**Comments to the Author**

1. Is the manuscript technically sound, and do the data support the conclusions?

Reviewer #1: Yes

Reviewer #2: Yes

2. Has the statistical analysis been performed appropriately and rigorously? 

Reviewer #1: Yes

Reviewer #2: Yes

3. Have the authors made all data underlying the findings in their manuscript fully available?

Reviewer #1: Yes

Reviewer #2: Yes

4. Is the manuscript presented in an intelligible fashion and written in standard English?

Reviewer #1: Yes

Reviewer #2: Yes

5. Review Comments to the Author

Reviewer #1: Dear Authors

This study evaluates the fine and meso-scale movements patterns of two species of pelagic dolphin. The work is well structured and just minor revisions are needed. I recommend it be published after considering these minor suggestions

Introduction

113-119. This paragraph is repeated (lines: 101-107), please delete it.

Material and methods

328: Why 'our second goal'? No specification about it was mentioned in the introduction (lines 173-176) and it is a bit confusing. Delete 'our second goal' or rewriting the goals in the introduction section.

361: Replace 'Generalized Linear Mixed Models' by 'GLMM'

368: Replace 'maximum likelihood 'by 'ML'

385: Change the number of the figure. The sentence corresponds to figure 5

604: Add 'CRW' after 'correlated random walk'

610. Replace'correlated Random walk' by 'CRW'

Results

415: In Figure 9, it appears that differences in step length become more pronounced for groups of 50 or more common dolphins. Why are only groups larger than 100 mentioned?

Discussion

Lines 521-523 and 539-541: These sentences are very similar to each other and repeat results (Line 439-440). I suggest rewriting them.

Tables

Table 1: In the 'Total' row, what do the values of 14 and 18 indicate for the mean number of locations for common and dusky dolphins, respectively? The sum of the monthly means/12 gives other values. Specify in the legend

Table 3 could be move to supplementary material

Figure legends

Figure 1:Specify that the square indicates the area where the study was conducted.

Figure 2, line 875: Delete 'empty' (the grey circles in the figure appear to be filled in).

Add the description of the behavioral state category "other" used in Figure 6 to the rest of the figures that have behavior as a variable (2,3,4,7 and 9).

Reviewer #2: Overall, this is an interesting study, with some novel (although expected) information about the movement patterns of two species of dolphins, in two different areas. Even though there are some limitations to the methodology (for example, the fact that it is based on observational data, meaning that the data is only diurnal, and collected for a limited number of hours), it can be replicated in studies where the populations are more studied, and included as an additional methodology.

Detailed review:

Title: add "species", after the word dolphins

Page 2, line 29: correct the word “effectiveness”

Page 2, line 37: it should be explained what the results for the common dolphin were. As it is: "did not" it can mean a lot of things.

Page 2, line 38: which dolphins? Both species? If so, it must be clear.

Page 2, line 40: "replace “these dolphins” by "both species of dolphins"

Page 3, lines 42-45: This paragraph should be replaced because it's confusing as it is.

Page 5, line 84: replace “and” by "or", as I think the authors are giving examples.

Page 5, line 89: add "e.g."

Page 9, line 166: I suggest replacing “search” by "foraging"

Page 9, line 175: did the authors mean "the variability of prey in the field"?

Page 12, line 236: Do the authors mean that "sampling periods" were larger? If so, it should be replaced, because as it is it seems that the authors encounter larger groups in the summer months.

Page 12, line 246: it is possible, although very difficult. Maybe replace by "...within a group is very difficult without artificial or natural marks."

Page 13, line 257-258: delete “at the beginning of the tracking”

Page 18, line 370 (Results): It seems that general results are missing. It would be good to have a general idea of the total effort and sightings for both species. Can the authors provide more information? Although the number of sightings is in table 1, it is not clear what the general effort was.

Page 18, line 385: Suggestion: replace by "noticeable";

Page 18-19, line 385-386: The sentence is confusing, rephrase for better understanding.

Page 19, line 389: delete “a”

Page 19, line 406: delete “as”

Page 20, line 427: Missing a parenthesis.

Page 23, line 485: replace “dolphins” for "species"

Page 24, line 507: feeding or foraging? It is expected that when they are feeding they will follow the preys' movements, while when they are foraging it should be the dolphins decision.

Page 25, line 522: correct “straightforward”

Page 25, line 537: delete “during winter”

Page 25, line 541: delete “also”

Page 26, lines 549-553: this is not always the case. Although equally invasive, deploying satellite and d-tags, doesn't imply capturing or manipulating the animals. The authors should rephrase.

Page 26, line 553: But has the restriction of not being able to follow the animals all the time. This should be addressed.

Page 27, line 567: delete “do” and replace "not leave" by” not to leave"

6. PLOS authors have the option to publish the peer review history of their article (what does this mean?). If published, this will include your full peer review and any attached files.

Reviewer #1: No

Reviewer #2: No

---

## [Author Response · Author response to Decision Letter 0]

4 Oct 2022

Dear Editor

David Hyrenbach, Ph.D.

Academic Editor

PLOS ONE

We are sending you the revised version of the ms:

PONE-D-22-19932

Seasonal variation and group size affect movement patterns of two pelagic dolphins (Lagenorhynchus obscurus and Delphinus delphis)

Within this letter you will find our response to each of your comments and how we addressed each one, as well as reviewers requirements and observations.

We expect they are stated clearly and referenced in a way they are easily find and checked in the text.

We expect the ms is now well suited for its publication in PlosOne.

Below we copied your letter and responded point by point.

Thanks in advance

Silvana Dans

Dear Dr. Dans,

Thank you for submitting your manuscript to PLOS ONE. After careful consideration, we feel that it has merit but does not fully meet PLOS ONE’s publication criteria as it currently stands. Therefore, we invite you to submit a revised version of the manuscript that addresses the points raised during the review process.

In addition to the specific questions and edits suggested by the two reviewers, please make sure to address these comments:

Please make sure you address the reviewer requests for clarifications of the text, and for re-writes of certain sections.

Response: We are very thankful for reviewers comments and suggestions, which improved the ms.

In Figure 9, the differences in step length become more pronounced for groups of 50 or more common dolphins. Is this the case? Do the analyses support this assertion? If this pattern was not addressed, can you test it now? 

Otherwise, why are only groups larger than 100 mentioned in the text?

Response: Yes larger groups performed longer steps, but depending on the species (species*group size), then common dolphins larger groups performed longer steps, while different dusky dolphin groups sizes performed similar step length. 

The effect of the 3-way interaction term behavior*species*group size was not tested, as none of the remaining 3-way interactions. But looking at figure 9, predicted curves for dusky dolphin group sizes are more overlapped, although different between feeding and traveling. In the case of common dolphin, curves are more separated among group sizes, then the degree of overlapping comparing feeding and traveling, is higher. However, the curves do not overlap looking at larger common dolphin groups, and this is more evident for the last size category which is >100 animals. 

We removed the sentence “however, in common dolphin groups larger than 100 individuals, it is possible that the difference between traveling and feeding step length is as pronounced or more than in the dusky dolphin”, because this paragraph was dealing with behavior and step length. The relationship between step length and group size is presented in the next paragraph, and it was confusing here.

Therefore the whole paragraph from lines 454-469 was rewritten:

“Group size had an important effect on step length, since two interaction terms including the group size significantly improved the model. In general, larger groups performed longer steps, but the relationship depends on the species and behavior, since these interaction terms were both significant (group size*species and group size*behavior in table 4). Larger dusky dolphin groups did longer steps, as described previously, but steps length is quite similar among different group sizes, compared to common dolphins—the predicted curves of step length are more separated among different common dolphin group sizes, and more overlapped among dusky dolphin group sizes (fig 9). This tendency to perform larger steps when groups are larger also depends on the behavior; particularly larger groups perform longer steps when they are traveling than when they are feeding. Although the 3-ways interaction term was not tested, predicted curves for dusky dolphin group sizes are more overlapped, although different between feeding and traveling. In the case of common dolphin, curves are more separated among group sizes, then the degree of overlapping comparing feeding and traveling, is higher. However, the curves do not overlap looking at larger common dolphin groups, and this is more evident for the last size category which is >100 animals (fig 9). “

In Table 1, what do the values of 14 and 18 indicate for the mean number of locations for common and dusky dolphins, respectively? The sum of the monthly means / 12 gives other values.

Response: The values in each row correspond to monthly means and the value in the row “total” corresponds to the overall mean. It is explained in the table caption

Table 3 could be moved to the supplementary material section.

Done.

Response: We are sending all the requested files through editorialmanager site and figures were uploaded through PACE

I also included the details of the institutions and awards to be considered in the financial disclosure, in the cover letter.

We look forward to receiving your revised manuscript.

Kind regards,

David Hyrenbach, Ph.D.

Academic Editor

PLOS ONE

Journal Requirements:

Response: We checked for styles and formatting

2. Please ensure to move your statement about permits from the acknowledgment section to the methods section of the manuscript: "This work was carried out under permits of the Dirección de Fauna y Flora Silvestre and Ministerio de Turismo de la Provincia de Chubut and complies with the current laws of Argentina.

DONE

Response: We moved “Fieldwork was also supported by Hydrosport SRL and Asociación Civil Cota Cero Buceo” to acknowledgments section since they were not grants.

4. Please expand the acronym “PNUD ARG” (as indicated in your financial disclosure) so that it states the name of your funders in full.

Response: It was expanded to Programa de las Naciones Unidas para el Desarrollo Argentina

"NO authors have competing interests." 

Response this information was included in the cover letter

Response: We recently published the R script as well as data in github. The DOI number is available now:

https://doi.org/10.5281/zenodo.7117362

7. We note that Figure 1 in your submission contain [map/satellite] images which may be copyrighted. All PLOS content is published under the Creative Commons Attribution License (CC BY 4.0), which means that the manuscript, images, and Supporting Information files will be freely available online, and any third party is permitted to access, download, copy, distribute, and use these materials in any way, even commercially, with proper attribution. For these reasons, we cannot publish previously copyrighted maps or satellite images created using proprietary data, such as Google software (Google Maps, Street View, and Earth). For more information, see our copyright guidelines: http://journals.plos.org/plosone/s/licenses-and-copyright.

Response: the base map was constructed by free software QGis, and it was used as a base map. Additionally another elements were added to the base map. The figure as presented here is original.

a) You may seek permission from the original copyright holder of Figure 1 to publish the content specifically under the CC BY 4.0 license. 

8. Please ensure that you refer to Figure 5 in your text as, if accepted, production will need this reference to link the reader to the figure.

DONE

DONE

Response: references were checked

Reviewers' comments:

Reviewer's Responses to Questions

Comments to the Author

1. Is the manuscript technically sound, and do the data support the conclusions?

Reviewer #1: Yes

Reviewer #2: Yes

2. Has the statistical analysis been performed appropriately and rigorously? 

Reviewer #1: Yes

Reviewer #2: Yes

3. Have the authors made all data underlying the findings in their manuscript fully available?

Reviewer #1: Yes

Reviewer #2: Yes

4. Is the manuscript presented in an intelligible fashion and written in standard English?

Reviewer #1: Yes

Reviewer #2: Yes

5. Review Comments to the Author

Reviewer #1: Dear Authors

This study evaluates the fine and meso-scale movements patterns of two species of pelagic dolphin. The work is well structured and just minor revisions are needed. I recommend it be published after considering these minor suggestions

Response: We thank the reviewer for their nice comments.

Introduction

113-119. This paragraph is repeated (lines: 101-107), please delete it.

Done.

Material and methods

328: Why 'our second goal'? No specification about it was mentioned in the introduction (lines 173-176) and it is a bit confusing. Delete 'our second goal' or rewriting the goals in the introduction section.

Response: It was a mistake. Thank you for spotting it. We removed that. 

361: Replace 'Generalized Linear Mixed Models' by 'GLMM'

DONE

368: Replace 'maximum likelihood 'by 'ML'

DONE

385: Change the number of the figure. The sentence corresponds to figure 5

DONE

604: Add 'CRW' after 'correlated random walk'

DONE

610. Replace'correlated Random walk' by 'CRW'

DONE

Results

415: In Figure 9, it appears that differences in step length become more pronounced for groups of 50 or more common dolphins. Why are only groups larger than 100 mentioned?

Response: The idea of this sentence was misunderstood, and it was removed. The effect of group size on step length, and the effect of the interaction between group size and behavior and group size and species was well, are presented in the next paragraph, lines 454-469. See comments to the editor´s requirements.

Discussion

Lines 521-523 and 539-541: These sentences are very similar to each other and repeat results (Line 439-440). I suggest rewriting them.

Response: The paragraph was rewritten. It now reads: 

about seasonal patterns, both dolphins moved with shorter steps during winter, and longer steps during summer. Also, the season has a significant effect on the step length depending on the turning angle (Fig 9). This means that when dolphins do not change the direction, which is indicated by values around 0 in the distribution of turning angles, they will move a longer distance. Therefore, in summer, dolphins move longer distances in a more straightforward movement, while in winter, the movement is more erratic and moving shorter distances. The seasonal variation of the relationship between step length and turning angle of dusky dolphins’ movement in Nuevo Gulf may be explained in terms of the abundance and distribution of anchovy shoals. There is some evidence about a seasonal change in foraging behavior of dusky dolphins inhabiting Nuevo Gulf [33] (Degrati et al. 2012). Dusky dolphins were observed in coordinated diving apparently in a feeding activity, contrasting to the surface feeding observed during summer. This change was suggested to be related to a change in the location of anchovy shoals, since both anchovies and dusky dolphins were found in deeper waters during the cold season [34] (Degrati et al. 2013). Although these results come from only one year, they suggest a seasonal pattern in prey distribution. More recently, a study about abundance and distribution of pelagic ensembles, including anchovies and red lobsters Munida gregaria, carried out in both gulfs during 2016-2018, indicated seasonal changes in the depth, the morphology and the energetic density of anchovy shoals [66] (Luzenti 2022). In Nuevo Gulf, during winter, the number of shoals per unit area and the energetic density are higher during winter than during summer. However, shoals are smaller and separated by shorter distances during winter than in summer [66] (Luzenti 2022). However, the seasonal variation of movement pattern of common dolphins in San Matias Gulf is more difficult to explain. In San Matias Gulf, shoals were also smaller in winter, but the number of shoals and the energetic density was lower [66] (Luzenti 2022). 

Tables

Table 1: In the 'Total' row, what do the values of 14 and 18 indicate for the mean number of locations for common and dusky dolphins, respectively? The sum of the monthly means/12 gives other values. Specify in the legend

Response: The values 14 and 18 correspond to the grand means, therefore these values are different from doing the sum of the monthly means/12.

It was specified in the legend that “total” row corresponds to the grand mean

Table 3 could be move to supplementary material

Done.

Figure legends

Figure 1:Specify that the square indicates the area where the study was conducted.

DONE

Figure 2, line 875: Delete 'empty' (the grey circles in the figure appear to be filled in).

DONE

Add the description of the behavioral state category "other" used in Figure 6 to the rest of the figures that have behavior as a variable (2,3,4,7 and 9).

DONE

Reviewer #2: Overall, this is an interesting study, with some novel (although expected) information about the movement patterns of two species of dolphins, in two different areas. Even though there are some limitations to the methodology (for example, the fact that it is based on observational data, meaning that the data is only diurnal, and collected for a limited number of hours), it can be replicated in studies where the populations are more studied, and included as an additional methodology.

Response: We thank the reviewer for their positive comment.

Detailed review:

Title: add "species", after the word dolphins DONE

Page 2, line 29: correct the word “effectiveness”DONE

Page 2, line 37: it should be explained what the results for the common dolphin were. As it is: "did not" it can mean a lot of things. DONE

Page 2, line 38: which dolphins? Both species? If so, it must be clear. 

Response: Both species, it was rewritten (now in line 41)

Page 2, line 40: "replace “these dolphins” by "both species of dolphins"DONE

Page 3, lines 42-45: This paragraph should be replaced because it's confusing as it is. 

Response: It was replaced by:

However, dusky dolphins rely on anchovy to a larger extent than common dolphins. In Nuevo Gulf, anchovy shoals are smaller and separated by shorter distances in winter and dusky dolphins´ movement pattern is consistent with this. 

Page 5, line 84: replace “and” by "or", as I think the authors are giving examples. The reviewer is right. DONE

Page 5, line 89: add "e.g."DONE

Page 9, line 166: I suggest replacing “search” by "foraging" 

Response: We preferred to keep “search” instead of “foraging” to be consistent with the general movement theoretical framework, though for marine pelagic predators “search” is mostly with the aim to “forage”. 

Page 9, line 175: did the authors mean "the variability of prey in the field"?

Response: By the variability of prey field we mean temporal as well as spatial variability of prey. The prey field refers to the structure and distribution of prey biomass across the seascape. It was added in the text line 89.

Page 12, line 236: Do the authors mean that "sampling periods" were larger? If so, it should be replaced, because as it is it seems that the authors encounter larger groups in the summer months. The reviewer is right.

 Response: We replaced it.

Page 12, line 246: it is possible, although very difficult. Maybe replace by "...within a group is very difficult without artificial or natural marks."DONE.

Page 13, line 257-258: delete “at the beginning of the tracking”DELETED

Page 18, line 370 (Results): It seems that general results are missing. It would be good to have a general idea of the total effort and sightings for both species. Can the authors provide more information? Although the number of sightings is in table 1, it is not clear what the general effort was.

Response: We added the requested information as a new “SURVEY EFFORT” (IN DAYS) column in Table 1.

Page 18, line 385: Suggestion: replace by "noticeable";DONE

Page 18-19, line 385-386: The sentence is confusing, rephrase for better understanding.

DONE. The sentence now reads: 

“…and maximum values of step length were recorded when dolphins were traveling (Fig 6).”

Page 19, line 389: delete “a”DONE

Page 19, line 406: delete “as”DONE

Page 20, line 427: Missing a parenthesis.DONE

Page 23, line 485: replace “dolphins” for "species"DONE

Page 24, line 507: feeding or foraging? It is expected that when they are feeding they will follow the preys' movements, while when they are foraging it should be the dolphins decision.

Response: The behavioral state was defined as feeding. Foraging is wider than feeding, and may include other behavioral states, like traveling from one shoal to another.

Page 25, line 522: correct “straightforward”DONE

Page 25, line 537: delete “during winter”DONE

Page 25, line 541: delete “also”DONE

Page 26, lines 549-553: this is not always the case. Although equally invasive, deploying satellite and d-tags, doesn't imply capturing or manipulating the animals. The authors should rephrase.

Response: It was rephrased as follows:

“ tagging and tracking dolphins in the wild, usually, but not always, requires the capture, manipulation and release of animals.”

Page 26, line 553: but has the restriction of not being able to follow the animals all the time. This should be addressed.

Response: Addressed. The following lines were added:

“Although it has the restriction of not being able to follow the animals all the time, species studied in the present work feed mainly during daylight hours therefore the observed movement pattern would represent most of their foraging strategy at a fine scale.”

Page 27, line 567: delete “do” and replace "not leave" by” not to leave"

DONE

---

## [Editor Report · Decision Letter 1]

11 Oct 2022

Seasonal variation and group size affect movement patterns of two pelagic dolphin species (Lagenorhynchus obscurus and Delphinus delphis)

PONE-D-22-19932R1

Dear Dr. Dans,

We’re pleased to inform you that your manuscript has been judged scientifically suitable for publication and will be formally accepted for publication once it meets all outstanding technical requirements.  Thank you for addressing all the comments of the reviewers and the editor, and or providing the requested information about funding and potential conflicts of interest.

Kind regards,

David Hyrenbach, Ph.D.

Academic Editor

PLOS ONE
---

## [Editor Report · Acceptance letter]

18 Oct 2022

PONE-D-22-19932R1 

Seasonal variation and group size affect movement patterns of two pelagic dolphin species (Lagenorhynchus obscurus and Delphinus delphis) 

Dear Dr. Dans:

I'm pleased to inform you that your manuscript has been deemed suitable for publication in PLOS ONE. Congratulations! Your manuscript is now with our production department. 

Kind regards, 

on behalf of

Dr. David Hyrenbach 

Academic Editor

PLOS ONE